# Low Distortion Block-Resampling with Spatially Stochastic Networks

**Sarah Jane Hong**[*]
Latent Space
sarah@latentspace.co

**Martin Arjovsky**[* †]
École Normale Supérieure
martinarjovsky@gmail.com

**Darryl Barnhart**[*]
Latent Space
darryl@latentspace.co

**Ian Thompson**[*]
Latent Space
ian@latentspace.co

## Abstract

We formalize and attack the problem of generating new images from old ones that are as diverse as possible, only allowing them to change without restrictions in certain parts of the image while remaining globally consistent. This encompasses the typical situation found in generative modelling, where we are happy with parts of the generated data, but would like to resample others ("I like this generated castle overall, but this tower looks unrealistic, I would like a new one"). In order to attack this problem we build from the best conditional and unconditional generative models to introduce a new network architecture, training procedure, and a new algorithm for resampling parts of the image as desired.

## 1 Introduction

Many computer vision problems can be phrased as conditional or unconditional image generation. This includes super-resolution, colorization, and semantic image synthesis among others. However, current techniques for these problems lack a mechanism for fine-grained control of the generation. More precisely, even if we like certain parts of a generated image but not others, we are forced to decide on either keeping the generated image as-is, or generating an entirely new one from scratch. In this work we aim to obtain a generative model and an algorithm that allow for us to resample images while keeping selected parts as close as possible to the original one, but freely changing others in a diverse manner while keeping global consistency.

To make things more precise, let us consider the problem of conditional image generation, where the data follows an unknown distribution $\mathbb{P}(x, y)$ and we want to learn a fast mechanism for sampling $y \in \mathcal{Y}$ given $x \in \mathcal{X}$. The unconditional generation case can be instantiated by simply setting $x = 0$. The current state of the art algorithms for image generation usually employ generative adversarial networks (GANs) [15, 28, 18] when presented with a dataset of pairs $(x, y)$. Conditional GANs learn a function $g_\theta : \mathcal{Z} \times \mathcal{X} \rightarrow \mathcal{Y}$, and afterwards images $\hat{y}$ are generated from $x$ by sampling $z \sim P(z)$ and outputting $\hat{y} := g_\theta(z, x)$. The distribution $P(z)$ is usually a fixed Gaussian distribution, and the GAN procedure makes it so that $g_\theta(z, x)$ when $z \sim P(z)$ approximates $\mathbb{P}(y|x)$ in a very particular sense (see [15, 3] for more details). As such, GANs create a diverse set of outputs for any given $x$ by transforming the $z$'s to different complex images.

One limitation of the above setup is that given a generated sample $\hat{y} = g(z, x)$, we are restricted to accept it and use it as-is for whatever our downstream task is, or generate an entirely new sample by

---

[*]Equal contribution.
[†]Work performed while at Latent Space.

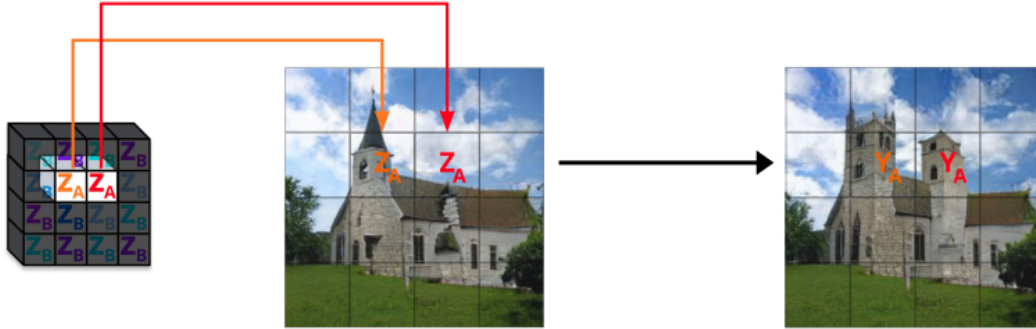

Figure 1: A diagram of Spatially Stochastic Networks. We decompose the latent code $z$ spatially into independent blocks, and regularize the model so that local changes in $z$ correspond to localized changes in the image. We then resample parts in the image by resampling their corresponding $z$'s.

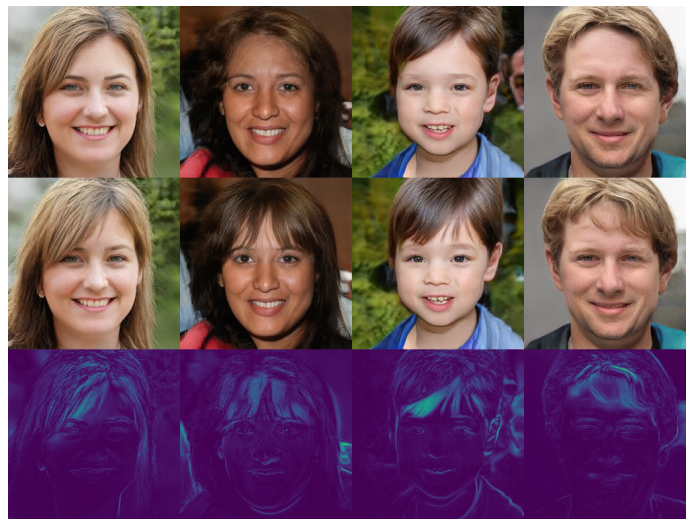

Figure 2: Resampling a person's hair. The top row consists of unmodified generations of our models, Spatially Stochastic Networks (SSNs), trained on FFHQ [17]. With SSNs, resampling two $z$'s near the top of each persons head makes spatially localized changes (middle row) while also allowing for minimal necessary changes in other parts of the image (third row), unlike in traditional inpainting.

resampling $z' \sim P(z)$ and obtaining $\hat{y}' = g(z', x)$. There is no in-between, which is not optimal for many use cases.

Consider however the case of Figure 2. Here, we have a GAN trained to do unconditional image generation, and the generations (top row) are of high-quality. However, we would like to provide the user with the ability to modify the hair in the picture while leaving the rest unchanged. In essence, instead of regenerating the entire image, we would like to keep some parts of the image we are happy with as much as possible, and only resample certain groups of pixels that correspond to parts we are unhappy with. The task here is image generation, but it could be super resolution, colorization, or any task where spatially disentangled resampling would be useful.

Our solution to this task is simple: we split the latent code $z$ into many independent blocks, and regularize the generator so that each block affects only a particular part of the image. In order to achieve good performance, we need to make architectural and algorithmic changes drawing from the best conditional and unconditional generative models. This solution, called Spatially Stochastic Networks (SSNs), is schematized in Figure 1. In the second row of Figure 2 we can see that we successfully achieve the resampling of the hair, while minimally affecting the rest of the image.

While much work has been done in *inpainting*, which consists of resampling parts of the image while leaving the rest *exactly* fixed, in problems with structured data this limits drastically the diversity of

the resampling. For instance, if we wanted to inpaint a set of pixels corresponding to the hair of a person, we would need to leave the rest of the face exactly fixed. It is unlikely that a resampling of the hair can be achieved without changing even minimally the facial structure and keeping a globally consistent image. We would also need a mask that tells us exactly where every single hair pixel is located, which is usually unavailable. However, with our new ideas, we can select a large block of pixels containing hair and the resample will change those pixels while *minimally* affecting the rest of the image. Another example of this is seen in Figure 1, where we only roughly select the blocks of pixels containing a tower and other pixels not in those blocks need to modified in order for changes to render a consistent resampling. Thus, in order to obtain diverse new resamplings that minimally change the rest of the image, we need to allow a *small* distortion in other parts of the image. This is what we understand as *Low Distortion Block-Resampling*, or LDBR.

The contributions of this paper are as follows:

- In section 2 we introduce a mathematical framework to study the low distortion block-resampling problem and showcase how it relates to other problems in computer vision such as inpainting.

- In section 3 we study why current techniques are unsuited to solve the LDBR problem. From this analysis, we construct Spatially Stochastic Networks (SSNs), an algorithm for image generation directly designed to attack this problem. In the process, we introduce several new developments for spatially-conditioned generative adversarial networks, which are of independent interest.

- In section 4 we perform both qualitative and qualitative experiments showing the workings and excellent performance of SSNs.

- In section 5 and section 6 we relate SSNs to other works, and conclude by posing open problems and new research directions that stem from this work.

## 2 Low Distortion Block-Resampling

Let $y \in \mathbb{R}^{n_y \times n_y \times 3}$ be an RGB image. We define a *block* simply as a subimage of $y$. More concretely, let $I = \{1, \ldots, n_y\}$, and $J_1, \ldots, J_{n_{\text{blocks}}} \subseteq I \times I$ be disjoint subsets of indices such that $\cup_{a=1}^{n_{\text{blocks}}} J_a = I$. Then, the block with index $a$ is defined as $y_a := (y_{i,j,1}, y_{i,j,2}, y_{i,j,3})_{(i,j) \in J_a}$ where $y_{i,j,1}, y_{i,j,2}, y_{i,j,3}$ are the red, green and blue intensity values for pixel $(i, j)$ respectively. We will often refer to both $y_a$ and $a$ as blocks when the meaning is obvious from the context. While in this paper we will focus mainly on rectangular (and in particular square) blocks with the form $J_a = \{i, \cdots, i+l\} \times \{j, \cdots, j+l'\}$, all our techniques and ideas translate to non-rectangular subimages unless we make explicit mention of it.

The goal of resampling block $a$ can be informally stated as: given a pair $(x, y)$ from $\mathbb{P}$, generate an alternative $y'$ via a stochastic process $P^a(y'|(x, y))$ such that all blocks $b$ different than $a$ are preserved as much as possible (i.e. $y_b' \approx y_b$ for all $b \neq a$), and such that if we resample every block (i.e. consecutively apply $P^a$ for all $a$), we arrive to an image $y^*$ whose distribution is $\mathbb{P}(y|x)$. To summarize, we want to construct a new plausible image such that only one block is allowed to change unrestricted at a time, and such that resampling every block constitutes resampling the whole image.

**Definition 1** *Let* $\{P^a(y'|x, y)\}_{a=1,\ldots,n_{\text{blocks}}}$ *be a set of conditional probability distributions over* $\mathcal{Y}$*, one for each block* $a = 1, \ldots, n_{\text{blocks}}$*. We say that* $\{P^a\}_{a=1,\ldots,n_{\text{blocks}}}$ *is a* block-resampling *of the probability distribution* $\mathbb{P}(y|x)$ *if when* $y^{(n_{\text{blocks}})}$ *is constructed by the sequential sampling process*

$$y^{(0)} \sim \mathbb{P}(\cdot|x)$$
$$y^{(1)} \sim P^{a_1}(\cdot|x, y^{(0)})$$
$$y^{(2)} \sim P^{a_2}(\cdot|x, y^{(1)})$$
$$\cdots$$
$$y^* := y^{(n_{\text{blocks}})} \sim P^{a_{n_{\text{blocks}}}}(\cdot|x, y^{(n_{\text{blocks}}-1)})$$

*we have that the distribution of* $y^*$ *is* $\mathbb{P}(y|x)$*.*

*In words, if we start from a sample $y^{(0)}$ of $\mathbb{P}$ and we resample every block in an arbitrary order, we obtain a new independent sample from $\mathbb{P}$.*

Note that simply setting $P^a(\cdot|x,y) = \mathbb{P}(\cdot|x)$ gives a trivial resampling for $\mathbb{P}$, which simply resamples the entire image every time. This, however, collides with our goal of each time resampling an individual block while leaving the other blocks as untethered as possible. This is exactly why we need a *low distortion* block resampling, which we now define.

Let $D : \mathbb{R}^{J_a \times 3} \times \in \mathbb{R}^{J_a \times 3} \to \mathbb{R}_{\geq 0}$ be a notion of distortion between subimages such as the Euclidean distance between pixels or the Earth Mover's distance[31]. Then, we define the problem of low distortion block resampling as the constrained optimization problem

$$\min_{P^a(y'|x,y)} \quad \mathbb{E}_{(x,y)\sim\mathbb{P}}\left[\sum_{a=1}^{n_{\text{blocks}}} \mathbb{E}_{y'\sim P^a(\cdot|x,y)}\left[\sum_{b\neq a} D(y_b, y'_b)\right]\right] \qquad \text{(LDBR)}$$

$$\text{subject to} \quad \{P^a\}_{a=1,\ldots,n_{\text{blocks}}} \text{ is a block-resampling of } \mathbb{P}$$

At this point, it is important to clarify the distinction between resampling and *inpainting* (see for instance [9]). Inpainting constitutes the goal of sampling from the conditional probability distribution $\mathbb{P}(y'_a|x, (y_b)_{b\neq a})$, so resampling the block $y_a$ conditioned on $x$ and the other blocks $y_b$, which are held *exactly fixed*.[3] In LDBR we allow $y'_b$ to differ from $y_b$, but want to enforce that resampling all blocks constitutes a resampling of the entire image. However, sequentially inpainting all the different blocks in general does not constitute a resampling of the entire image. If it did, then inpainting would give a solution of (LDBR) with 0 distortion, which in general does not have to exist. Consider the simplistic example in which $y$ has only two pixels $y_0$ and $y_1$, each of which is a separate $1 \times 1$ block. If $\mathbb{P}(y = (1,1)|x) = \mathbb{P}(y = (0,0)|x) = 1/2$ for some $x$, then sequentially inpainting starting on $y = (1,1), x$ would do nothing, since $\mathbb{P}(y_0 = 1|y_1 = 1, x) = 1 = \mathbb{P}(y_1 = 1|y_0 = 1, x)$. In particular, one could never attain $y' = (0,0)$ by this process starting with $y = (1,1)$. In fact, the only way that sequential inpainting can yield a block-resampling is if blocks are independent to each other conditioned on $x$ (something virtually impossible for structured data). This is due to the fact that after sequential inpainting, $y^{(1)}$ has distribution $\mathbb{P}(y'|x, (y_b^{(0)})_{b\neq a_1})$ which, unless blocks are independent conditioned on $x$, is different to $\mathbb{P}(y|x)$, and since $y_{a_1}^{(1)} = y_{a_1}^{(n_{\text{blocks}})}$, we get that $y^{(n_{\text{blocks}})}$ cannot have distribution $\mathbb{P}(y|x)$, thus failing to be a block-resampling for $\mathbb{P}$.

As mentioned, current generative adversarial networks are unsuited to solve the (LDBR) problem, since the only mechanism to generate new samples they have is to resample an entire image. In the next section we introduce Spatially Stochastic Networks, or SSNs, a particular kind of conditional GANs paired with a new loss function, both specifically designed to attack the (LDBR) problem.

## 3 Spatially Stochastic Networks

As mentioned, conditional GANs currently offer one sampling mechanism given an input $x$: sample $z \sim P_Z(z)$ and output $\hat{y} = g(x, z)$. Our idea to attack problem (LDBR) is simple in nature: split $z$ into blocks, and regularize the generator so that each latent block $z_a$ minimally affects all image blocks $y_b$ for $b \neq a$. Therefore, by consecutively resampling all individual latent blocks $z_a$, we obtain an entire resampling of the image $y$. In the case where blocks are just rectangular parts of the image, $z$ becomes a 3D spatial tensor. We then need a generator architecture that performs well when conditioned on a spatial $z$, and it needs to be regularized so for any given block $z_a$, it affects as much as possible only the image block $y_a$. We call the combination of these two approaches *Spatially Stochastic Networks* or SSNs, which we can see diagrammed in Figure 3.

More formally, if we define $P(\hat{y}|x)$ is the distribution of $g(x, z)$ with $z \sim P_Z(z)$ and $P_Z$ be such that $z_a$ and $z_b$ are independent for all $z \neq b$ (such as $P_Z = \mathcal{N}(0, I)$). Then, given $\hat{y} = g(x, z)$, let $P^a(\hat{y}'|x, \hat{y})$ be defined as the distribution of $\hat{y}' = g(x, \tilde{z})$ where $\tilde{z}_a = z_a$, and $\tilde{z}_b = z'_b$ for $b \neq a$ and

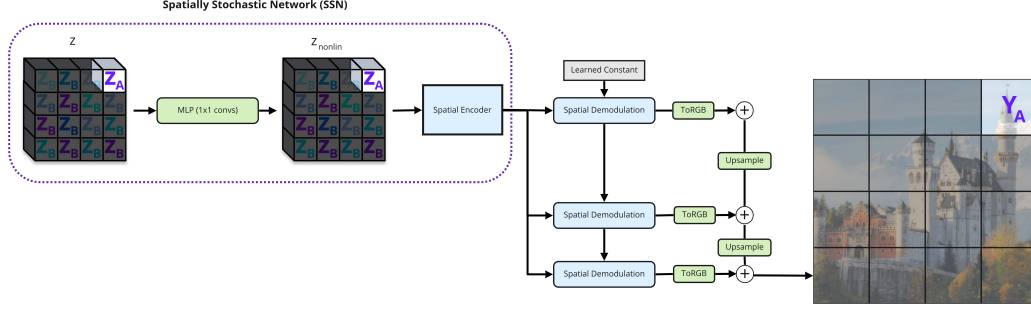

Figure 3: Spatially Stochastic Networks. Each block $z_a$ is a vector $z_a \in \mathbb{R}^{n_z}$. If we have $n_{\text{blocks}} = n_w \times n_h$, then $z \in \mathbb{R}^{n_w \times n_h \times n_z}$. The generator is regularized so that each $z_a$ affects mostly $y_a$.

$z, z'$ independent samples of $P_Z$. It is trivial to see that $(P^a)_{a=1,\ldots,n_{\text{blocks}}}$ is a resampling of $P(\hat{y}|x)$, since applying $P^a$ consecutively just consists of taking a new independent $z \sim P(z)$. We can see this illustrated in Figure 3: if we resample $z_a$ for all $a$, this just amounts to sampling a new $z$, and hence a new independent sample from the generator.

As mentioned, for this approach to succeed we require two things: we need the generator distribution $P(\hat{y}|x)$ to be similar to the data distribution $\mathbb{P}(y|x)$, and we need the resampling of $P(\hat{y}|x)$ described above to have low distortion. For the first objective, we need to come up with an architecture for the generator and training regime that achieves the best possible performance when conditioned on a spatial $z$. We achieve this goal in subsection 3.1. For the second objective of the resampling having low distortion, we need a regularization mechanism to penalize $z_a$ from affecting other blocks $y_b$ with $a \neq b$, which we study in subsection 3.3.

We begin with the design of a generator architecture that maximizes performance when conditioned on spatial $z$. To do so, we leverage ideas from the best conditional and unconditional generative models, as well as introduce new techniques.

## 3.1 Spatial Conditioning Revisited

The best current generator architecture and training regime for spatially conditioned generators is (to the best of our knowledge) SPADE [28]. While SPADE was a major improvement over previous methods for spatially conditioned generative modelling, its performance still lags behind from the best of unconditional generation methods like StyleGAN2 [18]. In addition to the performance and quality benefits, StyleGAN2 uses a simpler training process than SPADE. In particular, it doesn't need the additional auxiliary losses of SPADE (which require training a separate VAE). In this section, we adapt the spatial conditioning elements of SPADE to work with the techniques of StyleGAN2, creating a new model for spatially conditioned GANs. When used with a spatial $z$, we will show this model performs on par with StyleGAN2, whose quality far surpasses that of SPADE.

One of the most important aspects of this contribution is the observation that SPADE's conditioning has analogous downsides to those of the first StyleGAN [17]. In particularly, both models exhibit prominent 'droplet' artifacts in their generations (see Figure 4 left). The reason for these artifacts in StyleGAN is the type of conditioning from $z$ the model employs [18], which shares important properties with SPADE's conditioning. This problem of StyleGAN was solved in [18] by the introduction of normalizing based on expected statistics rather than concrete feature statistics for their conditioning layers. Following the same line of attack, we apply a similar analysis to the SPADE layers but whose normalization is based on expected statistics, thus eliminating the droplet artifacts from SPADE and yielding a new layer for spatial conditioning which we call *Spatially Modulated Convolution*.

$$\text{SpatiallyModulatedConv}_w(\mathbf{h}, \mathbf{s}) = \frac{w * (\mathbf{s} \odot \mathbf{h})}{\sigma_E(w, \mathbf{s})} \tag{1}$$

with

$$\sigma_E(w, \mathbf{s})^2_{c'} := \frac{1}{HW} \sum_{i=1}^{H} \sum_{j=1}^{W} \left( w^2 * \mathbf{s}^2 \right)_{c', i, j}$$

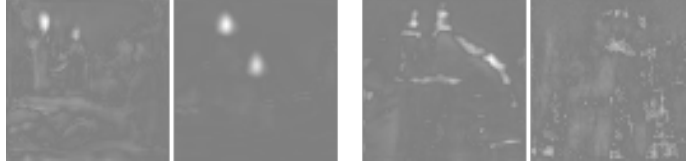

Figure 4: Left: droplet artifacts on the panel when using SPADE. Right: no droplet artifacts after introducing spatially modulated convolutions in SSNs. Images generated with the procedure of [18].

where $\mathbf{s} \in \mathbb{R}^{1 \times C \times H \times W}$ is the conditioning and $\mathbf{h} \in \mathbb{R}^{N \times C \times H \times W}$ is the input to the layer. Due to space constraints, we leave the full derivation of our new layer to Appendix A.

Now, the whole reason why we introduced spatially modulated convolutions is to avoid the droplet artifacts appearing in SPADE and thus get better quality generations when conditioning on spatial inputs. As can be seen in Figure 4, we successfully achieved the desired results: replacing SPADE layers with spatially modulated layers, we can see that droplet artifacts disappear.

Since the focus of our paper is on low distortion block resampling, we leave the application of spatially modulated convolutions for conditional image generation tasks like semantic image synthesis for future work. Given the drastic increase in performance from StyleGAN (which shares a lot of similarities with SPADE) to StyleGAN2 (of which one of the main changes is the adoption of modulated convolutions), we conjecture that there is a lot to be gained in that direction.

## 3.2 Leveraging Unsupervised Techniques

While our spatially modulated convolution got rid of bubble artifacts, there are a few other improvements introduced by StyleGAN2 that we can take advantage of to get the best possible performance and make the training process as simple as possible. First, we remove the VAE and the perceptual losses used in SPADE, thus reducing a lot of the complexity of the training process. Second, we utilize StyeGAN2's idea of passing $z$ through a nonlinear transformation to another latent code (which we call $z^{\mathrm{nonlin}}$) before passing it to the modulated convolutions. The way we do this is we apply the same MLP to each of the blocks $z_a$ to generate the blocks $z_a^{\mathrm{nonlin}}$. We implement this efficiently with $1 \times 1$ convolutions applied to $z$ directly. We also utilize skip connections, the general architecture, and the $R_1$ and path length regularization (with weights of 1 and 2 respectively) of [18]. A diagram of the final architecture, which we call Spatially Stochastic Networks, can be seen in Figure 3.

## 3.3 Low Distortion Regularization

The current architecture is well suited to employ a spatial noise, and hence it is easy to resample individual blocks of $z$. However, nothing in the loss function is telling the model that this resampling should have low distortion. In particular, no part of the loss encourages the generator so that changing $z_a$ minimally changes $y_b$ for $b \neq a$. We attack this problem by regularizing distortion explicitly.

Let $\tilde{z}^a(z, z')$ be the noise vector with block $a$ equal to $z_a$, and block $b$ equal to $z'_b$ for all $b \neq a$ (see Figure 3). Then, we can regularize directly for the distortion of the resampling.

$$R_D(g) := \mathbb{E}_{(x,y) \sim \mathbb{P}} \left[ \sum_a \sum_{b \neq a} \mathbb{E}_{z,z' \sim P_Z(z)} \left[ D\left(g(x,z)_b, g(x, \tilde{z}^a(z,z'))_b\right) \right] \right] \quad (2)$$

Equation (2) is just the cost of equation (LDBR) rewritten employing the reparameterization trick[19] over $P^a$. This way we explicitly encourage the model to induce a low distortion block resampling.

We also experimented with replacing the path length regularization term of [18] with one more explicitly designed for the LDBR setup without success. We leave these details to Appendix B.

## 3.4 Transfer Learning For High Resolution Experiments

In order to experiment at high resolutions, we take advantage of pretrained StyleGAN2 models. The reason for this is simple: experimenting at high resolutions from scratch simply has a prohibitive cost

| Configuration | FFHQ (256x256 pixels) | | | LSUN Churches (256x256 pixels) | | |
|---|---|---|---|---|---|---|
| | FID | PPL | Resampling | FID | PPL | Resampling |
| A Baseline StyleGAN2[18] | 19.76 | **137.33** | ✗ | **3.65** | 340.72 | ✗ |
| B SSNs | **12.24** | 151.01 | ✓ | 8.68 | **282.75** | ✓ |

Table 1: Comparison of StyleGAN2 and SSNs without distortion regularization. Lower scores are better for FID and PPL. Both models attain comparable quality, while SSN allows for block resampling.

for us, aside from being quite harmful to the environment. Before explaining our transfer protocol, it is good to justify its use with concrete numbers. All of the experiments in this paper used transfer. To give some perspective, training a single StyleGAN2 model from scratch on LSUN churches takes 781 GPU hours on V100s, which has a cost of about $2,343 USD, and 70.29 kilograms of $CO_2$ emitted into the atmosphere [20]. Using transfer, we only need 4 GPU hours, which translates to roughly $12 USD and only 0.36 kgs of $CO_2$. In total, all the experiments needed for this paper (including debugging runs and hyperparameter sweeps) had a cost of about $2,000 USD, and without transfer this would have required around $400,000 USD to run (incurring in almost 20,000 kgs of $CO_2$).

Our transfer protocol is as follows. First, we copy all the weights and biases directly from pretrained StyleGAN2 models ([1] for LSUN and [2] for FFHQ) that correspond to analogous components: we map the weights from the 8-layer MLP from the original StyleGAN2 to an 8-layer set of 1x1 convolutions in SSNs, the weights from the StyleConvs from StyleGAN2 are mapped to the corresponding weights in the SpatialDemod blocks in SSNs, and finally, the ToRGB blocks in StyleGAN2 are mapped to the ToRGB coming out of spatial demod in SSNs. Our spatial encoder module has no direct analogy in StyleGAN2, so the layers in the spatial encoder are randomly initialized.

# 4    Experiments

We experiment with the FFHQ [18] faces and the LSUN churches [39] datasets at a resolution of $256 \times 256$ pixels. The latent code has dimension $z \in \mathbb{R}^{4 \times 4 \times 512}$ for SSN and $z \in \mathbb{R}^{1 \times 1 \times 512}$) as per StyleGAN2's default configuration. We provide both quantitative and qualitative experiments. The quantitative ones have as a purpose to study what is the trade-off between quality of the generations and distortion, and also provide guidelines for selecting the hyperparameter that balances between these quantities. The qualitative ones are meant to show what these numbers mean visually. In particular, we will see that in both these datasets we can achieve close to optimal quality (in comparison to the best model available) and visually interesting resamplings, including those of the form "I like this generated church overall, but this tower looks unrealistic, I would like a new one".

As a sanity check, we first compare the performance of unregularized SSNs with that of StyleGAN2, the current state of the art in unsupervised generative modelling. This is meant to verify that we don't lose performance by introducing a spatial $z$ and the spatially modulated convolutions, which are necessary for our end goal of resampling. We can see these results in table Table 1, where we indeed observe no noticeable loss in quality.

Second, we study the trade-off between quality and low distortion. This is determined by the regularization parameter for the term (2), which we call $\lambda_D$. To study this, we ablate different values of $\lambda_D$ for the FFHQ dataset, which can be seen in table Table 2. Based on these results, we chose the hyperparameter of $\lambda_D = 100$ for our qualitative experiments, since it gave a reduction in distortion of an entire order of magnitude while only incurring a minor loss in FID (note that the FID with $\lambda_D = 100$ is still marginally better than that of the original StyleGAN2). We also plot the corresponding Pareto curve in Figure 5 in the Appendix. It is important to comment that these curves are arguably necessary for comparing different solutions to the (LDBR) problem, since different algorithms are likely to incurr in different tradeoffs of quality and distortion.

## 4.1    Qualitative Experiments

In Figure 6 of Appendix C we show several resamples in LSUN churches. We can see that the images are of high quality, and the changes are mostly localized. We are able to see towers appearing, structural changes in the buildings, or even trees disappearing. Furthermore, in some of the cases the

| Configuration | FFHQ (256x256 pixels) | | |
|---|---|---|---|
| | FID | PPL | Distortion |
| Baseline StyleGAN2 | 19.76 | 137 | N/A |
| SSNs, $\lambda_D = 0$ (no distortion reg.) | 12.24 | 151 | 0.028 |
| SSNs, $\lambda_D = 1$ | 13.47 | 154 | 0.028 |
| SSNs, $\lambda_D = 10$ | 12.80 | 130 | 0.017 |
| SSNs, $\lambda_D = 100$ | 15.24 | 83 | 0.0043 |
| SSNs, $\lambda_D = 1000$ | 66.74 | 75 | 0.0004 |
| SSNs, $\lambda_D = 10000$ | 128.99 | 55 | 0.0001 |

Table 2: Ablation for different strengths of the low distortion regularization weight $\lambda_D$. Lower is better for both FID and PPL (quality metrics) and for distortion. The value of $\lambda_D = 100$ achieves a significant reduction in distortion without incurring a significant loss in quality (strictly better in both FID and PPL than the state of the art StyleGAN2 baseline). Surprisingly, the PPL metric decreases as the regularization strength increases.

resampled area is of relatively poor quality while the resample is not (and vice versa), thus allowing for resampling to serve as a refining procedure. Similar changes in FFHQ can be seen in Figure 7 in Appendix C with changes in glasses, eye color, hair style, among others. In most of the images, we also see small changes outside the resampled blocks which are needed to keep global consistency, something that couldn't happen with inpainting (see Figure 8 of Appendix C for more details).

Before we conclude and highlight the many avenues for future work, we first discuss how this relates to other works in the literature.

## 5 Related Work

Now that we have explored resampling in the context of SSNs, it is worth revisiting how related work has interpreted various forms of resampling, and how this compares to or complements our approach. The relevant literature for manipulating the latent space of GANs can largely be partitioned into two major categories: methods that focus on manipulating global attributes such as age and gender, and methods that focus on making localized changes associated with segmentation maps and/or instance maps.

The first category involves approaches that aim to manipulate global attributes of an agnostic decoder's latent space. For example, GANSpace [16] applies PCA to the latent space or feature space of a decoder to modify global attributes like the make of a car, background, or age. [32] similarly manipulates global attributes showing the latent can be disentangled after linear transformations, or [22] by performing optimization in the latent space of a decoder. However, these methods need many optimization steps for a single encoding or feature. Other methods train an encoder to factorize the latent in specific ways without modifying the generator, such as in [25], [40], or [14]. Another class of approaches to manipulating global attributes trains an encoder jointly with the decoder, as done in ALAE [30], ALI [13], and BigBiGAN [12]. As it stands, most of the encoder based methods involve changing the entire image, but there is nothing fundamentally blocking extending them to support LDBR.

The second category involves modulating features via prior assumptions about which semantic features would be useful to modulate, such as faces in [24], [34], [29], or that there is only a single central object in each image as in [10], [23], [33]. SSNs are in this category, aiming to modify local regions. If high resolution segmentation maps are available, the promising work of Bau et al.'s [7], [5], and [35], [4], [38] have shown that it is possible to not only make geometric changes in the image by changing the segmentation maps, but also modify the textural properties within a given segmentation instance or class. Beyond being practical, the interpretable factorization in this line of work builds on similar approaches to understand individual units in classifiers in [6], [26], [27], [11], and provides insight into how these models are capturing or not capturing the distribution. The mode dropping phenomena highlighted by these segmentation-based approaches inspired our resampling work, particularly Bau et al.'s demonstration of how under-capacity GANs drop difficult classes such as humans in front of buildings from the support of the distribution [8]. In the conditional literature there have been several improvements in leveraging even stronger priors, whether temporal as in [36] spatial as in [23], or making better use of high resolution segmentation information as in [21].

However, we also believe that local resampling without explicit segmentations or other strong priors can be useful. For example, often segmentations are not available, or the strong geometric prior of segmentation may be too restrictive. When coupled with segmentation, the latent representation tends to capture primarily textural details, whereas in our approach with SSNs the latent representation also captures geometric detail (it is more flexible at the cost of being less precisely controllable compared to segmentation approaches). In summary, our technique is useful when one does not have semantic segmentation available, or one wants to try out significant geometric changes not constrained by segmentation maps. This enables the resampling of semantically higher level structures like towers, hair, and glasses to make changes that are both geometric and textural.

## 6    Conclusions and Future Work

We have shown that generative model outputs can be modified in an incremental and well-defined way, with appropriate regularization. We also combine spatial conditioning with unconditional image generation using state-of-the-art architectures.

This reframing of the inpainting problem opens up a number of new lines of work. First, the use of MSE in pixel space as a distortion metric is a priori a terrible choice, with the Earth Mover's distance or MSE on feature spaces being semantically more meaningful notions of distortion. Second, we have not explored the use of spatially modulated convolutions and StyleGAN2-like ideas in spatial conditioning tasks like semantic image synthesis. Given the vast quality difference from SPADE (the current state of the art on these tasks) to StyleGAN2, it is likely that there is still room for significant improvement. Third, while we have focused on resamplings at one resolution dominated by the dimensions of $z$, one could think of having multiple 3D $z$'s operating at different scales, thus providing finer control.

Finally, in this work we have focused on pre-specified rectangular blocks that come simply by putting a grid in the image. However, we could think of non-rectangular blocks that come from other parts of a computer vision pipeline itself. For instance, blocks could correspond to regions in a semantic segmentation map of the image (created either by a human or by a machine learning algorithm), and hence resampling said blocks would constitute a resampling of objects in the picture. We are particularly excited in this direction, which could help open a vast amount of possibilities in terms of content creation and modification.

## 7    Broader Impact

The main goal of this paper is to give the user of a generative model finer control of its samples. This can have positive outcomes in the use case of creative applications of GANs, such as design, art, and gaming. Particularly, when the user is not the developer of the technology (for instance, it can be a player in a game who wishes to create a new level), we aim for him or her to be able create without being hindered by technical requirements.

Currently, developers and artists need expensive skills, experience, and separate tools to produce content. This has a negative downstream impact on the diversity of content that is ultimately produced. Representation is not equal, as content skews towards representing those who can afford to become developers. We believe creativity is evenly distributed across location, race, and gender. Techniques like SSNs that make content creation more accessible can help bridge this gap in representation.

Furthermore, any technique that is based on learning from data is subject to the biases in the training distribution. We believe resampling approaches like SSNs can help to visualize and understand these biases.

As any technology that promises to give easier access, it has the potential for misuse. One could imagine cases where generative models are used to create things that may be harmful to society, and this can lower the technical entry barrier to misusers of this technology. For instance, SSNs could be applied towards harmful DeepFakes. We thus believe that it's our duty as researchers to participate in the discussion of regulating these technologies so that they can be guided towards positive outcomes.

## Footnotes

[3]Sometimes inpainting is defined slightly differently [37]: given a pair $(y, x) \sim \mathbb{P}$ and access to $x, (y_b)_{b\neq a}$, come up with $y'(x, (y_b)_{b\neq a})$ that minimizes the expected mean squared loss (or cross-entropy) to $y$. It is easy to see that the optimal solution is $\arg\min_{y'(x,(y_b)_{b\neq a})} \mathbb{E}_{(x,y)\sim\mathbb{P}}\|y'(x, (y_b)_{b\neq a}) - y\| = \mathbb{E}[y'|x, (y_b)_{b\neq a}]$, therefore this definition of inpainting amounts to returning the mean of the above conditional distribution $\mathbb{P}(y'_a|x, (y_b)_{b\neq a})$ rather than sampling from it, in which case the rest of the analysis remains the same.

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
