[Supplementary Material]

# A Spatial Conditioning Without Bubble Artifacts

Let us begin by recalling how SPADE works, and study where its defects come from. SPADE is based on the utilization of Spatially Adaptive Normalization (SPADE) layers, which given an input $\mathbf{h} \in \mathbb{R}^{N \times C \times H \times W}$ and *spatial* conditioning 'style' $\mathbf{s}_s, \mathbf{s}_b \in \mathbb{R}^{1 \times C \times H \times W}$

$$\text{SPADE}(\mathbf{h}, \mathbf{s}) = \mathbf{s}_s \odot \frac{\mathbf{h} - \mu_{\mathrm{D}}(\mathbf{h})}{\sigma_{\mathrm{D}}(\mathbf{h})} + \mathbf{s}_b \qquad (3)$$

where $\odot$ means pointwise multiplication and $\mu_{\mathrm{D}}(\mathbf{h}), \sigma_{\mathrm{D}}(\mathbf{h}) \in \mathbb{R}^{1 \times C \times 1 \times 1}$ are per-channel statistics of $\mathbf{h}$:

$$\mu_{\mathrm{D}}(\mathbf{h})_c := \frac{1}{NHW} \sum_{n=1}^{N} \sum_{i=1}^{H} \sum_{j=1}^{W} \mathbf{h}_{n,c,i,j} \qquad (4)$$

$$\sigma_{\mathrm{D}}(\mathbf{h})_c^2 := \frac{1}{NHW} \sum_{n=1}^{N} \sum_{i=1}^{H} \sum_{j=1}^{W} \left( \mathbf{h}_{n,c,i,j} - \mu_{\mathrm{D}}(\mathbf{h})_c \right)^2 \qquad (5)$$

These statistics are calculated via averages over examples and all spatial dimensions. To clarify, the subtraction and division in (3) are broadcasted on non-channel dimensions, and the pointwise multiplication and addition are broadcasted over examples.

SPADE layers are remarkably similar to the Adaptive Instance Normalization (AdaIN) layers that are used in StyleGAN to condition on $z$. In StyleGAN, the authors have $z \sim P_z$, and first obtain $s = (s_s, s_b) = F(z)$ with $s_s, s_b \in \mathbb{R}^{1 \times C \times 1 \times 1}$ and $F$ is a learned transformation from the noise vector $z$. Finally, the conditioning of the generator's output $y = g(z)$ (StyleGAN is an unconditional generative model) is done via AdaIN layers conditioned on $s(z)$. AdaIN layers are defined as

$$\text{AdaIN}(\mathbf{h}, s) = s_s \frac{\mathbf{h} - \mu(\mathbf{h})}{\sigma(\mathbf{h})} + s_b \qquad (6)$$

An important difference is that $\mu(\mathbf{h}), \sigma(\mathbf{h}) \in \mathbb{R}^{N \times C \times 1 \times 1}$ are not averaged over the data

$$\mu(\mathbf{h})_{n,c} := \frac{1}{HW} \sum_{i=1}^{H} \sum_{j=1}^{W} \mathbf{h}_{n,c,i,j} \qquad (7)$$

$$\sigma(\mathbf{h})_{n,c}^2 := \frac{1}{HW} \sum_{i=1}^{H} \sum_{j=1}^{W} \left( \mathbf{h}_{n,c,i,j} - \mu(\mathbf{h})_{n,c} \right)^2 \qquad (8)$$

and hence AdaIN is applied independently across examples.

AdaIN (6) and SPADE (3) are incredibly similar, with the only differences being the spatial conditioning and that SPADE averages over datapoints while AdaIN does not. As mentioned in [18], AdaIN is prominent to have droplet-like artifacts. In Figure 4, we can see that SPADE has these droplet artifacts as well. The solution presented in [18] was to take out the mean normalization, and replace the statistics $\mu(\mathbf{h}), \sigma(\mathbf{h})$ with *expected* statistics, assuming $\mathbf{h}$ are independent random variables with expectation 0 and standard deviation 1. When merging scaling conditioning with $s = F(z) \in \mathbb{R}^{1 \times C \times 1 \times 1}$, convolution with a weight vector $w$, and subsequent normalization, they arrive to the layer

$$\text{ModulatedConv}_w(\mathbf{h}, s) = \frac{w * (s\mathbf{h})}{\sigma_E(w, s)} \qquad (9)$$

where $\sigma_E(w, s) \in \mathbb{R}^{1 \times C \times 1 \times 1}$ is the expected standard deviation of $w * (s\mathbf{h})$ assuming $\mathbf{h}$ are independent variables with zero mean and unit variance

$$\sigma_E(w, s)_{c'}^2 = \mathbb{E}_{\mathbf{h}} \left[ \frac{1}{HW} \sum_{i=1}^{H} \sum_{j=1}^{W} \left( (w * (s\mathbf{h}))_{c',i,j} - \mathbb{E}_{\mathbf{h}} \left[ (w * (s\mathbf{h}))_{c',i,j} \right] \right)^2 \right] \qquad (10)$$

$$= \sum_{i=1}^{H} \sum_{j=1}^{W} \sum_{c=1}^{C} w_{c',c,i,j}^2 s_c^2 \qquad (11)$$

In the same way, we can derive a spatially modulated conv by merging *spatial* conditioning with $\mathbf{s} \in \mathbb{R}^{1 \times C \times H \times W}$, convolution with a weight vector $w$, and subsequent normalization based on expected statistics. Thus, we arrive to our Spatially Modulated Convolution layer

$$\text{SpatiallyModulatedConv}_w(\mathbf{h}, \mathbf{s}) = \frac{w * (\mathbf{s} \odot \mathbf{h})}{\sigma_E(w, \mathbf{s})}$$

and in this case we have $\sigma_E(w, \mathbf{s}) \in \mathbb{R}^{1 \times C \times 1 \times 1}$ is the expected standard deviation of $w * (\mathbf{s} \odot \mathbf{h})$, which after some algebraic manipulations we can see equates

$$\sigma_E(w, \mathbf{s})^2_{c'} = \mathbb{E}_{\mathbf{h}} \left[ \frac{1}{HW} \sum_{i=1}^{H} \sum_{j=1}^{W} \left( (w * (\mathbf{s} \odot \mathbf{h}))_{c',i,j} - \mathbb{E}_{\mathbf{h}} \left[ (w * (\mathbf{s} \odot \mathbf{h}))_{c',i,j} \right] \right)^2 \right] \quad (12)$$

$$= \frac{1}{HW} \sum_{i=1}^{H} \sum_{j=1}^{W} \left( \sum_{i'=1}^{H} \sum_{j'=1}^{W} \sum_{c=1}^{C} w^2_{c',c,i',j'} \mathbf{s}^2_{c,i+i',j+j'} \right) \quad (13)$$

$$= \frac{1}{HW} \sum_{i=1}^{H} \sum_{j=1}^{W} \left( w^2 * \mathbf{s}^2 \right)_{c',i,j} \quad (14)$$

where the squares in (14) are taken element-wise.

This new normalization layer has similarities and fundamental differences with that of StyleGAN2 [18]. An important difference between our spatially modulated convolution (1) and the modulated convolution of StyleGAN2 (9) is that (1) cannot be expressed as a convolution $\tilde{w} * \mathbf{h}$ with a new set of weights. While one can rewrite (9) as $\left( \frac{sw}{\sigma_E(w,s)} \right) * \mathbf{h}$, one cannot do the same thing with equation (1). This is due to the fact that one cannot commute the pointwise multiplication of (1) with the convolution. In essence, this means that when conditioning on spatial inputs, modulating the inputs is inherently different to modulating the weights, while in the non-spatial case these are equivalent.

## B  Negative Results

### B.1  Low Distortion Path Length Regularization

We identified one potential problem with the path length regularization technique introduced in [18]. Path length regularization drives the generator so that the Jacobian-vector product $\frac{\partial g(z,y)}{\partial z}^T y$ has constant norm for all directions $y \in \mathcal{Y}$ and all $z$. In particular, this regularization term encourages all parts of $z$ to affect all parts of $y$ with equal strength, which directly contradicts the fact that we want $z_a$ to minimally affect blocks $y_b$ with $a \neq b$. Therefore, we want to adapt the regularization technique so that $\frac{\partial g(z,y)_a}{\partial z_a}^T y_a$ has large and constant norm for all $z, y_a$, and $\frac{\partial g(z,y)_b}{\partial z_a}^T y_b$ has *small* and constant norm for all $z, y_b$. We tried to achieve this by replacing the path length regularization with

$$\mathbb{E}_{z,y \sim N(0,I)} \left[ \sum_a \left( \| \frac{\partial g(z,y)_a}{\partial z_a}^T y_a \|_2 - \gamma_+ \right)^2 + \sum_{b \neq a} \left( \| \frac{\partial g(z,y)_b}{\partial z_a}^T y_b \|_2 - \gamma_- \right)^2 \right] \quad (15)$$

where $\gamma_+ >> \gamma_-$. Note that if one had $\gamma_+ = \gamma_-$ then this would be exactly the path length regularization of [18]. Taking $\gamma_+ >> \gamma_-$ allows us to keep the stability properties of this regularization, but driving $g$ so that $z_a$ minimally affects $y_b$.

Despite the rationale behind this idea, we could not find settings where we noticed a decrease in distortion that was not accompanied by a drastic decrease in quality. In particular, we could not observe any noticeable benefit by replacing the path length regularization term of [18] with (15). We experimented with $\gamma_+ = 1$, $\gamma_- = 0.1$, and regularization weights for (15) to one of $\{200000, 20000, 2000, 200, 20, 2\}$ without a perceived increase of quality for any given distortion value.

## C  Supplemental Figures

Figure 5: Pareto curve visualizing the trade-off between quality (measured by FID) and distortion for SSNs trained in FFHQ at 256 x 256 resolution. Based on these results we chose to use $\lambda_D = 100$ for the qualitative experiments since it incurred a negligible loss in FID while drastically decreasing distortion.

Figure 6: Generations of our SSN model and corresponding resamplings. The model was trained on 256 x 256 LSUN churches. The latent code has dimension $z \in \mathbb{R}^{4 \times 4 \times 512}$ and the new images were obtained by resampling the latent blocks $z_{(1,1)}$ and $z_{(1,2)}$. We can see that the new images change mostly locally, with elements like towers appearing or disappearing, or trees changing. However, some minor changes are present in other parts of the image in order to keep global consistency, something that inpainting would not be able to do. The quality is comparable to that of StyleGAN2 [18].

Figure 7: Generations of our SSN model and corresponding resamplings. The model was trained on 256 x 256 FFHQ. The latent code has dimension $z \in \mathbb{R}^{4 \times 4 \times 512}$ and the new images were obtained by resampling the latent blocks $z_{(1,1)}$ and $z_{(1,2)}$. We can see that the new images change mostly locally, with changes corresponding to the hair, eye color, expressions, glasses, and other semantic elements. Some minor changes are present in non-resampled parts of the image in order to keep global consistency, something that inpainting would not be able to do. The quality is comparable to that of StyleGAN2 [18].

Figure 8: We include additional experiments to highlight two things: The distinction between inpainting and potential advantages in certain situations, and the workings of SSNs with new block resolutions. In these experiments, we switch from $4 \times 4$ blocks to $8 \times 8$ blocks to showcase a more granular resampling. We resample the blocks constituting to the left eye in a picture three times (zoom to view well). The left image for each pair of images is the original generated image and the right is after resampling. All resamplings occur at the same location. The left-most, single image shows the underlying latent code dimension $z \in \mathbb{R}^{8 \times 8 \times 512}$ overlayed onto the original generated image, with the blocks to be resampled highlighted in red. In A we obtain a local resampling: the left eye region is more lit and less shadowy. This is a typical desired case of LDBR. In B, we see change that spans outside the resampled region with glasses appearing across the face. This resampling adheres semantically since it would be out of distribution to have glasses appear only on one half of the face. In C we receive little to no change, which is also in distribution but arguably not the desired use case. The distinction between LDBR and inpainting is very clear in case B. Inpainting by definition is not allowed to make changes to the area specified as conditioning, which includes the *right* eye. However, for SSNs, the other eye can be changed with added glasses. This example highlights the intrinsic trade-off between having a faithful (and diverse) resampling of the data distribution and low distortion.