[Reviews · NeurIPS 2020]

Review 1

Summary and Contributions: The authors present a method for isolating in a conditional generative network for spatial regions of the image for the purpose of resampling some regions while leaving others as close as possible to unchanged.

Strengths: The presented approach is interesting and the theoretical exploration is clear, correct, and very illuminating. I especially appreciate the authors showing the relationship between the resampling of the blocks, the resampling of a whole image, and the conditional probability. As best I can confirm, this seems to be the first take on this approach. The authors have written on the potential significance of being able to resample in this flexible way.

Weaknesses: Based on an understanding that this work is not exactly meant to rely on copious empirical evaluation but that it is not so deeply theoretical either that it needs not be validated with experiments. I was disappointed in the experimental results presented on two axes. 1) Why isn’t the state of the art of inpainting contrasted against the approach? Especially, considering that it would be possible to replicate the same blocks and have side-by-side comparisons which would bring a lot of potential credence to the main claim that doing things in this way is necessary. 2) For a work of this type I would also like to see more exploration of the method within itself, namely, we have in the work a presentation of how the loss - hyperparameters effect quantitative measures but we do not have either qualitative or quantitative results based on the different selection of number of blocks, which effects the size of the blocks by extension.

Correctness: Other than the issues mentioned above, I could not find any errors or incorrect assertions.

Clarity: Yes the authors have presented it in an easy to understand way which does a good job of leading the reader through the approach. That said, I would prefer if the related work appeared in the more traditional place after the introduction especially in a case like this where the work is claimed to be strictly novel it would help to have a discussion of the justification to that end early to focus the reader.

Relation to Prior Work: This is another weakness of the work, the related work and set of references are surprisingly short. In addition, there is not much said about the wide array of other works that modify conditionally generated images by for example traversing the latent space along directions that have been trained to coincide with for instance gender, expression, and so on.

Reproducibility: No

Additional Feedback: After reading all the reviews, the rebuttal, and the discussion, I decided to update the rating. The authors have responded adequately in the rebuttal and the updated rating assumes that the authors will do a good job in making the necessary changes and additions to the final version of the paper.


Review 2

Summary and Contributions: This paper tries to modify part of the generated image while keeping the rest of the image unchanged. To achieve this, this paper proposed a new module named spatially stochastic networks. Both quantitative and qualitative results are reported in the paper.

Strengths: 1. Understanding and exploring current state-of-the-art generative models(StyleGAN2) is a very popular topic these days, also, modifying part of the generated image is a very interesting topic. 2. This paper proposes Spatially Stochastic Networks achieve Low Distortion Block-Resampling.

Weaknesses: 1. This paper doesn't do enough literature study, understanding and editing the generated images are widely studied in previous work like [1] and [2]. 2. The experimental results are not convincing, such as the results in Table 1. On the other hand, more qualitative results need to be presented to validate the effectiveness of the proposed framework. 3. The proposed SSNs are relatively too simple to achieve many applications like content-aware or user-customized element editing. [1] Bau D, Zhu J Y, Strobelt H, et al. Gan dissection: Visualizing and understanding generative adversarial networks[J]. arXiv preprint arXiv:1811.10597, 2018. [2] Bau D, Strobelt H, Peebles W, et al. Semantic photo manipulation with a generative image prior[J]. arXiv preprint arXiv:2005.07727, 2020.

Correctness: This paper doesn't do enough literature study on the research topic, many previous works try to solve this problem with proposed methods. So this work should compare their results with them and make claims based on these experimental results.

Clarity: Overall, this paper is clearly written. But this paper need more applications to validate the proposed framework.

Relation to Prior Work: This paper is lack of careful lirature study.

Reproducibility: Yes

Additional Feedback:


Review 3

Summary and Contributions: The paper formulates a quite unusual task which appears to be a flexible version of inpainting where we have a GAN-generated image and want to resample only some parts of the image. The rest of the image can be changed as well, but constrained to be close to the original. Most of architecture design is essentially upgrading SPADE with Style-GAN2 goodies. A major contribution is a new loss function that forces spatial structure on the latent code which makes it possible to resample only part of the image.

Strengths: It looks like the main strength is a new loss to regularize latent code and link it to the spatial structure of the image. It allows to do dynamic resampling of certain areas from GAN while keeping the other parts roughly the same. It seems to work quite well on visual samples in the paper and supplementary. It may have an impact on image generation software to allow more fine-grained user input rather than "accept/skip".

Weaknesses: In Table 1 it is clear that SSN performs much better than StyleGAN2 and it isn't explained why. It is quite surprising to me given that many improvements were motivated by StyleGAN2 and even most weights are directly used. Related work is very short. Figure 4 shows very low quality image and may scare some people away because of the quality concerns. As far as I remember Style-GAN2 paper shows droplet artifacts on otherwise good images.

Correctness: Yes, I think so.

Clarity: The paper is well-written. However, I found the notation very heavy, much heavier than it is needed for such a paper. Another thing I found confusing is x, y. In the literature I read x traditionally means image while y is a label. In this paper it is the other way around.

Relation to Prior Work: I think related work can be certainly extended. Most comparisons are made against inpainting and they are not really justified, just a short "inpainting can't do that".

Reproducibility: Yes

Additional Feedback: SGAN2 vs Style-GAN2 are used in the text inconsistently. It may confuse unfamiliar readers. I suggest to settle on Style-GAN2 name. I'd suggest to remove CO2 discussion. I believe it is quite misleading to attribute CO2 cost to NN training because it is not inherent to GPU computations per se. It stems from the fact that certain developed countries power certain datacenters with fossil fuels. It can be completely avoided by using nuclear energy or renewable sources. I hope the authors realize that even though SSN is a catchy acronym, it makes their work almost ungoogleable due to collision with a well-known American anachronism. I couldn't find information on the size of z. How many blocks were used in experiments? POST REBUTTAL: I have read the rebuttal and am fairly satisfied about the authors' responses.


Review 4

Summary and Contributions: The paper proposes a latent space for a styleGAN2+SPADE hybrid network that uses a latent code with a strong spatial block structure. In this way, they can resample parts of the image (i.e. change hair style) without being forced to hold all non-selected blocks strictly fixed (as is the case with inpainting.) The resulting network is well-described and shows good results.

Strengths: Overall I found the paper easy to follow. The network architecture is a natural combination of existing architectures, and I appreciated their selection/highlighting of the relevant parts of each network choice (ex. spatially modulated convolutions.) I think the problem formulation of altering spatial blocks is reasonable, although perhaps limiting in terms of other useful aspects of the styleGAN2 latent space. But the authors do a good job of pointing out other avenues in the future to expand on this such as using the semantic segmentation. Overall the results of the model and the quantiative evaluation are well conveyed and I think there is lots of interesting future work to build upon here.

Weaknesses: There are many consequences of the spatially-biased latent code that are not discussed. The most relevant being that if one has a global property (say, the hair color) that one wants to cycle, how easy is it to get the whole image to shift in this direction? Presumably, this would require consistently altering each subblock in some direction. As it is, while one might expect selecting all the hair tiles and resampling them to randomly resample the hair, this doesn't seem to be the case -- most of the alterations of this type in the paper and supplemental all seem to keep the hair color mostly fixed, but adjust/resample the orientation in some way. While this is interesting, showing what happens (perhaps artifacts etc.) when broader changes like this are resampled would be good to know. Similarly, many of the style-mixing properties of styleGAN aren't possible. If a low-error projection operation is possible, one could of course cycle latent codes between StyleGAN's latent code and the proposed latent decomposition to effect whatever changes are desired. In general more discussion and experiments here would noticably improve the paper. One other concrete thing is showing what happens when there are latent disagreements -- for exmaple, resampling the right half of the "hair tiles" to be blonde; what happens to the left half?

Correctness: Yes.

Clarity: Yes the paper was easy to read. Minor typos: "qualitative and qualitative experiments"

Relation to Prior Work: Yes prior work that I am aware of is discussed.

Reproducibility: Yes

Additional Feedback:

[Author Response · NeurIPS 2020]

We wanted to thank the reviewers for their thoughtful, concrete suggestions, which we have worked to address.

Reviewer #1 presents two main suggestions. First, an improved discussion and comparison with related work, particularly comparing with inpainting, and studying how our method relates to those based on exploring latent dimensions to generate new samples. Secondly, Reviewer #1 requested a more in-depth exploration of SSNs, and concretely suggested adding qualitative and quantitative results of the model when varying the size of the blocks. We agree that we should have done a better job with the literature review, and based on your comments we expanded the related work section in these directions. Furthermore, we added two experiments (see below), including the one you suggested with a different block size, and one that elucidates how inpainting and LDBR differ. We hope that this addresses some of your concerns, and if you think so, it would be great if you could update your score accordingly :).

To Reviewer #2, we apologize for missing the mentioned references! We have added these along with a detailed paragraph discussion (see the second to last paragraph in the added related work). We hope that this, and the rest of the added literature review and experiments, addresses some of your concerns and shows how we build on and complement existing works. If you think so, we would really appreciate it if you could update your score accordingly!

To Reviewers #3 and #4, thank you so much for your comments! We hope that the added literature review and experiments improved or solidified your opinions on the paper. We fixed all the mentioned typos and inconsistencies (e.g. now use exclusively the StyleGAN-2 name, added the detail of blocks being 4x4x512, and so forth). We also added an experiment very similar to the one suggested by Reviewer #4 (see below), so thank you for that!

**Added related work:** "The relevant literature for manipulating the latent space of GANs can largely be partitioned into two major categories: methods that focus on manipulating global attributes such as age and gender, and methods that focus on making localized changes associated with segmentation maps and/or instance maps.

The first category involves approaches that aim to manipulate global attributes of an agnostic decoder's latent space. For example, GANSpace (1) applies PCA to the latent space or feature space of a decoder to modify global attributes like the make of a car, background, or age. (2) similarly manipulates global attributes showing the latent can be disentangled after linear transformations, or (3) by performing optimization in the latent space of a decoder. However, these methods need many optimization steps for a single encoding or feature. Other methods train an encoder to factorize the latent in specific ways without modifying the generator, such as in (4), (5), or (6). Another class of approaches to manipulating global attributes trains an encoder jointly with the decoder, as done in ALAE (7), ALI (8), and BigBiGAN (9).

The second category involves modulating features via prior assumptions about which semantic features would be useful to modulate, such as faces in (10), (11), (12), or that there is only a single central object in each image as in (13), (14), (15). SSNs are in this category, aiming to modify local regions. If high resolution segmentation maps are available, the promising work of Bau et al (16), (17), and (18) (19), (20) have shown that it is possible to not only make geometric changes in the image by changing the segmentation maps, but also modify the textural properties within a given segmentation instance or class. Beyond being practical, the interpretable factorization in this line of work builds on similar approaches to understand individual units in classifiers in (21), (22), (23), (24), and provides insight into how these models are capturing or not capturing the distribution. The mode dropping phenomena highlighted by these segmentation-based approaches inspired our resampling work, particularly Bau et al.'s demonstration of how under-capacity GANs drop difficult classes such as humans in front of buildings from the support of the distribution (25).

We also believe that local resampling without explicit segmentations can be useful. For example, when segmentations are not available, or when the strong geometric prior of segmentation may be too restrictive. When coupled with segmentation, the latent representation tends to capture primarily textural details, whereas in our approach with SSNs the latent representation also captures geometric detail (it is more flexible at the cost of being less precisely controllable compared to segmentation approaches). In summary, our technique is useful when one does not have semantic segmentation available, or one wants to try out significant geometric changes not constrained by segmentation maps. "

**Added experiments:** We include additional experiments to highlight two things: The distinction between inpainting and potential advantages in certain situations, and the workings of SSNs with new block resolutions.

In these experiments, we switch from 4x4 blocks to 8x8 blocks to showcase a more granular resampling. We resample the blocks constituting to the left eye in a picture three times (zoom to view well).

In A we obtain a local resampling: the left eye region is more lit and less shadowy. This is a typical desired case of LDBR.

In B, we see change that spans outside the resampled region with glasses appearing across the face. This resampling adheres semantically since it would be out of distribution to have glasses appear only on one half of the

face. In C we receive little to no change, which is also in distribution but arguably not the desired use case.

The distinction between LDBR and inpainting is very clear in case B. Inpainting by definition is not allowed to make changes to the area specified as conditioning, which includes the *right* eye. However, for SSNs, the other eye can be changed with added glasses. This example highlights the intrinsic trade-off between having a faithful (and diverse) resampling of the data distribution and low distortion.

[Meta-Review · NeurIPS 2020]

Following the post-rebuttal discussion, the expert reviewers converged to a recommendation to accept the paper. The area chairs concur. The authors are encouraged to incorporate the additional experiments from the rebuttal into the final version.